# Assessing the Effect of Plant-Based Mince on Fullness and Post-Prandial Satiety in Healthy Male Subjects

**DOI:** 10.3390/nu14245326

**Published:** 2022-12-15

**Authors:** Beverly S. Muhlhausler, Damien Belobrajdic, Brooke Wymond, Bianca Benassi-Evans

**Affiliations:** 1Human Health, Health and Biosecurity, CSIRO, Adelaide, SA 5000, Australia; 2School of Agriculture, Food and Wine, The University of Adelaide, Adelaide, SA 5005, Australia; 3South Australian Health and Medical Research Institute, Adelaide, SA 5001, Australia

**Keywords:** plant-based meat, satiety, appetite, clinical trial

## Abstract

This study aimed to assess the effect of substituting plant-based mince for beef mince in a standard pasta meal on the amount consumed and on objective and subjective measures of post-prandial satiety. Healthy, adult males (*n* = 24) consumed a pasta lunch meal containing either plant-based or beef mince at separate visits, and the amount consumed measured at each visit. Perceptions of hunger, fullness and satisfaction were recorded and blood samples collected before and for 3 h after eating, when a buffet meal was provided. Participants consumed 586 kJ less of the pasta meal prepared with plant-based mince compared to beef mince (*p* < 0.05). Energy intake at the buffet meal and measures of fullness, satiety and satisfaction after the pasta meal were not different between plant and beef mince (*p* > 0.05). Post-prandial Glucagon-Like Peptide-1 (GLP-1), but not insulin or leptin concentrations, were lower after the plant-based pasta meal (*p* < 0.05). Our results suggest that the pasta meal containing plant-based mince was more satiating than an equivalent meal prepared with beef mince, and that this was not associated with greater energy intake at a subsequent meal occasion. Further studies that evaluate the longer-term effects of replacing meat with plant-based mince on energy intakes and explore the mechanisms underlying the lower consumption of the plant-based mince meal would be valuable.

## 1. Introduction

Growing concerns about the sustainability and environmental impacts of red meat consumption have resulted in increased uptake of plant-based red meat substitutes by consumers world-wide [1]. In addition to the environmental benefits of these products, they also have a markedly different nutritional composition in comparison to red meat. Some of these attributes, including higher sodium and saturated fat contents in some products have attracted criticism for their potential detrimental health effects. Conversely, others, particularly the higher fibre content, have led to suggestions that they have the potential to offer additional health benefits, especially given established health benefits of plant-based diets [1,2]. One such health benefit, and the focus of this study, is the potential for plant-based meats to produce a greater satiating effect in comparison to traditional meat products, and therefore potentially assist in supporting weight loss or weight maintenance in the longer term.

There are an increasing number of companies producing plant-based meats and have worked to create meat alternatives that look and taste like animal meat and can be used in cooking in the same way [1]. The nutritional profile of these plant-based meats varies markedly between products, but an increasing number have many nutritional attributes that are similar to meat, including comparable contents of macro- and micronutrients, including protein, iron, zinc, phosphorus and vitamins B12, B3 and B6. Despite these similarities, however, there are also key differences in the nutritional properties of plant-based meats compared to meat. This includes a higher fibre content and substantially lower moisture loss during cooking, which results in a lower energy density in the final cooked product [1]. Early consumer feedback has also suggested that plant-based meat products may also have a greater satiating effect, such that a lower quantity was required to satisfy hunger compared to an equivalent meal prepared with meat (personal communication, v2food^®^, Werribee, Australia). These anecdotal reports have since been supported by experimental studies [3,4], but further research is required to fully characterise this attribute, and in particular to determine whether any energy deficits at a single meal occasion are compensated for at subsequent meals.

While the mechanisms that underlie this potential increased satiating effect are not fully understood, it may be related to the higher fibre content and lower energy density of plant-based meat products, both of which are known to be associated with greater satiation effects [3]. By way of example, Williams and colleagues reported that participants felt less satiated when provided with a pasta meal with a higher energy density, and consequently consumed 153 ± 19 kcal more energy when compared to a similar pasta dish that was less energy-dense [5]. Collectively, this has led to the suggestion that substituting conventional meat products with plant-based products that are less energy-dense could result in less kilojoules being consumed at that specific meal occasion (satiation). In addition, it is important to confirm that this reduced energy intake at one meal is not compensated for by increased intake at subsequent meals, which would offset the benefits of any energy deficit at a single meal occasion.

The aim of this study, therefore, was to assess the effect of substituting plant-based mince for beef mince in a standard pasta meal on both the amount of the meal consumed and on feelings of fullness in the period after eating, as assessed by both subjective reporting, objective (biochemical) measures and the amount consumed at a subsequent meal occasion.

## 2. Materials and Methods

### 2.1. Study Population

Study participants were recruited over the period from the 13 July 2021 to the 7 September 2021. The eligibility criteria for inclusion were: healthy males aged between 18 and 56 years at the time of enrolment with a Body mass index (BMI) ≥ 18.5 and ≤27.5 kg/m^2^ and weight stable for at least 6 months (<±5 kg change). Participants needed to be consuming red meat regularly (at least 1× week) at the time of the study and be willing and able to attend the CSIRO nutrition clinic for around 8 h on two occasions across a two-week period.

Individuals were excluded from the study if they self-reported suffering from health conditions that could affect appetite/food intake or required a prescribed diet (e.g., gastrointestinal disease, type 1 or type 2 diabetes, cancer, renal or liver disease), chronic gastrointestinal symptoms (e.g., reflux, constipation) or a history of surgeries or use of medications known to potentially affect energy intake, appetite or gastrointestinal motor function (e.g., bariatric surgery or use of appetite-suppressants). Participants with known food allergies or intolerances to the study product, a history of eating disorders or drug/alcohol abuse and current smokers were also excluded. Individuals who had been involved in another trial/s within 28 days prior to Visit 1 involving the use of an investigational product that could potentially impact their ability to participate in the study or the study results were also ineligible.

### 2.2. Study Design

This was a single-blinded, randomised cross-over trial conducted at the CSIRO Nutrition and Health Research Clinic, South Australian Health and Medical Research Institute (SAHMRI), Adelaide. This study was conducted according to the guidelines described in the Declaration of Helsinki and all procedures involving human participants were approved by the CSIRO Human Research Ethics Committee (2021_046_HREC). Written informed consent was obtained from all participants.

### 2.3. Recruitment and Screening

The study was advertised through social media, and participants were instructed to contact CSIRO if they were interested in participating. Participants were recruited from the CSIRO Nutrition and Health Research database, and advertising through the CSIRO website and social media. Individuals who registered interest in participating in the study, and passed telephone screening, were invited to attend a screening appointment. Informed consent was conducted at the beginning of the screening appointment before any assessments were undertaken or further information was gathered.

At the screening appointment, height and weight were measured using the BSM370 Stadiometer to confirm self-reported BMI, and vital signs (temperature, blood pressure, heart rate and respiratory rate) were assessed. Details including participant’s age, gender, date of birth, ethnicity and contact details were collected by study staff. Self-reported medical and surgical history was obtained by a qualified nurse or suitably trained clinic staff member designee. The participant was questioned to assess the presence of any medical history that may impact their eligibility to participate in the study. Information was collected on all medications (prescription and non-prescription, vitamins, supplements and herbal medicines or investigational agents) taken by the participant 28 days prior to the screening appointment.

### 2.4. Randomisation

If participants met all the study criteria, they were then enrolled and randomised for the order in which they will receive the test meals. Randomised allocation was conducted according to an electronically generated randomisation plan that was created by an independent researcher. Participants and staff conducting the study appointments, measures and downstream analyses were blinded to group allocation.

### 2.5. Clinic Assessments

Participants attended 2 clinic appointments at least 1 week apart. Participants were instructed to consume their regular diet and abstain from any vigorous physical activity or alcohol for 24 h before attending each appointment and attended the clinic in the morning following an overnight fast. They were supplied with a standard breakfast meal at ~7 a.m., were provided with their test meal at lunch (~12 p.m.) and buffet meal 190 min after consuming the test meal (~3 p.m.). Participants remained in the clinic throughout the testing period. At each clinic visit, participants were questioned to assess if they had commenced, stopped or had any changes in concomitant medications since the previous visit.

### 2.6. Test Meals

The plant-based mince and beef mince were incorporated into a pasta bolognaise dish that was prepared in an accredited commercial kitchen. Plant-based mince was provided by v2food^®^ and beef mince was purchased from a local supermarket. Both mince types made up 45% of the total cooked meal weight. All ingredients used in the pasta dish were commercially available (garlic, olive oil, onion, tomato paste, passata, mixed herbs, cooking salt, black pepper, pasta shells, regular stock cube). The composition of both meals was assessed for the accurate calculation of nutrient intakes at the respective lunch meals (Table 1). The total caloric value of the pasta meal was similar between the plant-based and beef pasta meals (beef mince pasta meal, 788 kJ/100 g; plant-based mince pasta meal, 785 kJ/100 g). The test meals were manufactured in facilities certified to produce food for human consumption in accordance with an approved HACCP (Hazard Analysis Critical Control Point) plan. All the processing steps, from the purchase of the ingredients to the processing and handling conditions at each processing stage were approved by the CSIRO Food Risk Assessment Team (FRAT) team prior to conducting the study.

At each appointment, participants were provided with a pasta bolognaise meal at lunch time (~12 p.m.) that contained an equivalent amount of either beef mince or plant-based mince. Each meal contained 480 g mince and had a total weight of ~1.0 kg and participants were given 15 min to consume the meal and instructed to eat until they felt comfortably full. The amount of each meal consumed was measured by a member of the study staff and used to assess the weight and energy consumed by the participant.

### 2.7. Assessment of Post-Prandial Satiety

At 30, 15 and 5 min prior to consuming the test meal, and at 15, 30, 45, 60, 120 and 180 min after consuming the meal, perceptions of appetite and satiety were assessed using a validated visual analogue scale (VAS) [6]. At 190 min after consuming the test meals (~3 p.m.), subjects were presented with a standard cold buffet meal. The meal included sliced bread, chicken, ham, cheese, mayonnaise, custard, yogurt and fruit salad) and participants were allowed to eat *ad libitum* for up to 30 min until they were comfortably full. The amount of each item consumed was weighed and energy intake (kJ) and intakes of protein, carbohydrate, and fat were calculated using commercially available software (Foodworks 10; Xyris Software).

### 2.8. Assessment of Gut Comfort

Participants were asked to complete a Gastrointestinal Symptom Rating Scale, that asked about any gut symptoms they may have experienced for the previous 24-h, prior to consuming their breakfast and 24 h after consuming the lunch test meal at each study appointment.

### 2.9. Blood Collection

A temporary cannula was inserted ~30 min prior to the test meal to allow for the collection of regular blood samples before and after the test meal. Blood samples (~10 mL at each timepoint) were collected at 15 and 5 min before and 15, 30, 45, 60, 120 and 180 min after the consumption of the pasta meal. Blood was collected into vacutainers containing either a clot activator or K_2_-EDTA anticoagulant and proprietary additives. Serum tubes were left to clot at room temperature for up to 30 min and plasma tubes were stored on ice and centrifuged within 10 min of collection (Heraeus Multifuge X1R centrifuge; Thermo Scientific, Waltham, MA, USA). Plasma was centrifuged at 1900× *g* for 10 min at 4 °C and serum at 2800× *g* for 15 min at 4 °C. The resulting serum and plasma were separated into aliquots and stored at −80 °C within 30 min of collection for subsequent analysis of satiety and gut hormones (insulin, ghrelin and glucagon-like peptide-1 (GLP-1)).

### 2.10. Assessment of Metabolic and Gut-Hormones

Serum satiety hormone (insulin, ghrelin and GLP-1) concentrations were determined using commercial multiplex kits (EMD Millipore, Burlington, MA, USA). Samples and kit components were brought to room temperature before samples were mixed thoroughly and centrifuged at 9000× *g* for 3 min at room temperature using a benchtop microcentrifuge (Beckman Coulter microfuge 20 centrifuge, Brea, CA, USA). Commercial quality control (QC) samples were supplied with the kit and run on each plate. The kit was run according to kit instructions. Briefly, the plate was blocked with assay buffer prior to addition of standards, QCs and neat samples along with serum matrix, assay buffer and antibody-immobilized beads and incubated overnight at 4 °C in the dark on a plate shaker. After 16–18 h incubation, the plate was brought to room temperature and washed with wash buffer using an automatic platewasher (Biotek ELx405, BioTek Instruments Inc., Winooski, VT, USA) to remove unbound materials. Detection antibodies were then added and incubated briefly before the addition of streptavidin-phycoerythrin. After further incubation the plate was washed as previously described and sheath fluid added to the plate in preparation for reading on a Luminex 200 instrument (Luminex, Austin, TX, USA). The instrument was set up according to information provided in multiplex kit booklet. A JANUS robot (Perkin Elmer, Waltham, MA, USA) was used to prepare the standard curve and load plate with standards, QCs, samples, serum matrix and assay buffer. All other reagents were added by hand using a multichannel pipette.

### 2.11. Statistical Analysis

Statistical analyses were performed using GraphPad Prism (Version 9.0) and SPSS software (version 21; IBM, New York, NY, USA). Total weight of food consumed, energy intake and intake of specific nutrients was compared between the plant-based and beef pasta meals using a 2-tailed paired sample *t*-test. The same approach was used to compare energy and nutrient intake at the subsequent buffet meal and reported symptoms of gut comfort. Post-prandial subjective appetite measures and concentrations of metabolic and gut hormones were compared between the plant-based and beef clinic meals using a repeated measures ANOVA. Assumptions about normality and homogeneity of variance were checked graphically using residual plots and normal probability plots. Statistical significance was accepted at *p* < 0.05 and all data are presented as means ± SEs.

## 3. Results

### 3.1. Participant Flow

The participant flow for the study is shown in Figure 1. A total of 39 participants were screened, of whom 28 were eligible for the study and passed the screening appointment. These 28 participants were enrolled in the study of whom 24 completed all assessments.

### 3.2. Participant Characteristics

The mean age of the participants was 36.7 ± 2.0 years (range, 19–53 years) with an average weight of 76.2 ± 2.00 kg (range, 62–99 kg), mean height of 178.1 ± 1.4 cm (range, 167–191 cm) and mean BMI of 24.0 ± 0.4 kg/m^2^ (range, 21–27 kg/m^2^).

### 3.3. Food and Energy Intake of the Test Meal

The weight of the pasta meal consumed at the lunch meal was 72.1 g lower for the plant-based mince, compared to the beef mince (*p* < 0.05, paired *t*-test, Figure 2a). The amount of energy consumed at the meal was also significantly lower for the plant-based mince pasta meal compared to the beef mince pasta meal, with participants consuming an average of 586 less kJ of the plant-based pasta meal compared to the beef pasta meal (*p* < 0.05, paired *t*-test, Figure 2b).

Participants also consumed lower amounts of starch (beef mince,104.6 ± 6.4 g, plant-based mince, 88.8 ± 5.7 g; Mean Difference (MD), 15.8; *p* < 0.005) and fat (beef mince, 60.8 ± 3.7 g, plant-based mince, 54.6 ± 3.5 g; MD, 6.2 g; *p* < 0.05), and higher amounts of fibre (beef mince, 14.8 ± 0.9 g, plant-based mince, 30.5 ± 2.0 g; MD, 15.6 g; *p* < 0.001) when consuming the plant-based mince meal compared to the beef mince meal. There were no differences in the intakes of protein between the two test meals (beef mince, 86.9 ± 5.4 g, plant-based mince, 79.9 ± 5.1 g; *p* = ns).

### 3.4. Post-Prandial Satiety

There were no differences in the participants perceptions of hunger (Figure 3a), satisfaction (Figure 3b), fullness (Figure 3c) or how much they felt they could eat (Figure 3d) before or after eating between the plant-based mince and beef mince pasta meals, as assessed by a validated VAS. There was also no difference in the AUC for any of these measures between the plant-based and beef-mince pasta meals Hunger: beef meal, 6012 ± 467, plant-based meal, 5857 ± 519, *p* = 0.39; Satisfaction: beef meal, 13,476 ± 360, plant-based meal, 13,276 ± 394, *p* = 0.29; Fullness: beef meal, 13,528 ± 378, plant-based meal, 13,234 ± 401, *p* = 0.21; Desire to eat: beef meal, 8515 ± 579, plant-based meal, 7875 ± 466, *p* = 0.16).

### 3.5. Consumption at Buffet Meal

There was no difference in the amount of energy that participants consumed at the Buffet meal following the plant-based mince or beef mince pasta meal (beef mince, 3574 ± 330 kJ; plant-based mince, 3770 ± 398 kJ, *p* = 0.75). There were also no differences in the consumption of any individual nutrients at the buffet meal following the plant-based and beef pasta meals (Protein: beef meal, 43.9 ± 3.8 g, plant-based meal, 44.9 ± 4.3 g, *p* = 0.94; fat: beef meal, 30.2 ± 3.3 g, plant-based meal, 34.3 ± 4.0 g, *p* = 0.32; CHO, beef meal, 105.0 ± 10.4 g; plant-based meal, 100.2 ± 11.5 g, *p* = 0.92; Fibre, beef meal, 7.6 ± 0.8 g, plant-based meal, 7.3 ± 1.0 g, *p* = 0.70).

### 3.6. Post-Prandial Metabolic and Gut Hormones

Plasma insulin concentration increased following the consumption of the plant-based and beef pasta meals, reaching maximal levels 30 min after eating before declining and plateauing (F = 18.8, *p* < 0.001; Figure 4a). Conversely, ghrelin concentrations were highest prior to the consumption of the lunch meal and declined in the post-prandial period reaching nadir at 60 min (F = 14.1, *p* < 0.001; Figure 4b). There was no difference in the concentrations of these hormones between the plant-based and beef mince meals at any time before or after eating (insulin: F = 0.41, *p* < 0.001; ghrelin: F = 0.94, *p* = 0.48). Plasma GLP-1 concentrations were similar prior to the consumption of the beef and plant-based pasta meal but were lower at 60–120 min following the consumption of the plant-based pasta meal (Figure 4c, F = 3.91, *p* < 0.001). The AUC for plasma GLP-1 was also lower following consumption of the plant-based mince compared to the beef mince pasta meal (beef mince, 32,888 ± 2526 ng/mL/min, plant-based mince, 29,015 ± 1941 ng/mL/min, MD 3873 ng/mL/min, *p* < 0.001). The AUC for plasma insulin (beef mince, 265,901 ± 46,512 ng/mL/min, plant-based mince, 258,594 ± 51,390 ng/mL/min, *p* = 0.21) and ghrelin (beef mince, 7783 ± 1111 ng/mL/min, plant-based mince, 8048 ± 1211 ng/mL/min, *p* = 0.40) did not differ by meal type.

### 3.7. Gut Comfort

Gut comfort scores were generally close to 5 for the majority of symptoms assessed, indicating that few participants experienced any symptoms in the 24 h following consumption of the lunch meal. There were also no differences in gut symptoms in the 24 h following the consumption of the plant-based and beef pasta lunch meals (Table 2). No adverse events were reported.

## 4. Discussion

This study has demonstrated that food intake, both in terms of weight of the meal and energy intake, was significantly lower when healthy male participants were provided with a pasta meal containing plant-based mince, when compared to an equivalent pasta meal containing regular beef mince. This supports our hypothesis that participants would consume lower amounts of plant-based mince compared to regular mince when provided with *ad libitum* access. We also found that participants reported no differences in their hunger, fullness or satisfaction scores before or after consuming the meal and did not consume more food at a subsequent buffet meal, suggesting that the energy deficit at the lunch meal did not result in a decrease in post-prandial satiety or satisfaction, nor a compensatory increase in energy intake at the subsequent meal event. While the mechanisms underlying this greater short-term satiating effect of the plant-based meat is unclear, the maintenance of post-prandial GLP-1 concentrations after consuming the plant-based mince points to potential involvement of the gut endocrine system.

Given the rapid expansion of the plant-based ‘meat’ market globally over the past few years, there has been considerable interest in understanding the nutritional attributes of plant-based meat, and the impacts of consuming these products on a range of health outcomes. One area of interest has been the effects of these products on post-prandial satiety, with anecdotal reports suggesting a tendency for individuals to consume lower quantities of plant-based meats to achieve satiety compared to standard meat, something that is supported by the findings of this study. Our findings are also consistent with a similar study, which examined the effect of two energy and macronutrient-matched meals (a processed meat and cheese meal vs. a vegan meal containing tofu) on measures of post-prandial satiety and gut hormone concentrations in 3 groups of men (type 2 diabetic, obese and healthy). The authors reported that participants reported greater satiation after the vegan meal compared to the meat meal in all groups, supporting the suggestion that the plant-based meal was more satiating [4]. It is also in line with the results of a recent study which investigated the pattern of post-prandial amino acid release in individuals consuming a burrito prepared with either animal or plant-based mince, and found no difference in hunger, satisfaction or fullness scores in the 4 h post-consumption between meal types [7]. In this prior study, however, participants in all groups consumed the same amount of the test meals, rather than consuming until satiated. To our knowledge, there are no previous studies that have directly compared the intake of a plant-based vs. meat meal in individuals allowed to eat until they are full, making the current study an important addition to knowledge in the area of plant-based meats.

The mechanisms underlying the lower intake of the plant-based pasta meal compared to the beef pasta meal remain to be determined. One possibility is that it is related to the higher fibre content and/or lower energy density of plant-based mince compared to beef mince, both of which are known to be associated with greater satiation effects [3]. By way of example, Williams and colleagues reported that participants felt more satiated when provided with a pasta meal with a lower energy density, and consequently consumed 153 ± 19 kcal less energy when compared to a similar pasta dish that was more energy-dense [5] However, while fibre intake has been associated with increased post-prandial satiety in some studies, systematic reviews have suggested that this effect varies considerably between fibre types, and therefore it may not be possible to directly attribute the increased satiating effect of the plant-based product solely to its higher fibre content [8,9]. In the study cited above, in which healthy, type 2 diabetic and obese men consumed either a vegan or meat meal, the authors reported greater increases in gut hormones related to appetite control (in particular GLP-1 and amylin) following consumption of the vegan meal, suggesting that the greater satiating effects may be mediated by different effects of plant-based vs. beef mince on the post-prandial release of gut peptides [4]. This is somewhat at odds with the results of the current study, in which GLP-1 concentrations were lower following consumption of the plant-based compared to beef-mince pasta meal. The reason for this is unclear, particularly given that there was no difference in consumption at the subsequent meal occasion, at which point the differences in circulating GLP-1 concentrations were most marked. The effect does not appear to be mediated by difference in post-prandial insulin release or circulating ghrelin levels, all of which were not different between the two meal occasions. An alternate possibility is that the lower consumption of the plant-based mince was due to a lower liking of the taste and/or texture of this meal compared to the beef mince pasta meal. If this were the case, however, we would have expected to see more differences in the satiety post-prandial satiety scores between the groups. It would be interesting to include a Likert scale for each meal in any future studies, so that we could more accurately capture participants’ rating of how much they liked the taste, and whether this differed between the two test meals.

There have been suggestions in some studies that a high fibre load has the potential to result in adverse gut symptoms, particularly bloating and excessive gas [10,11]. The fibre intake in the plant-based pasta meal was in excess of 30 g, which represents a very high fibre load. It was therefore positive that there was no evidence of any adverse gastrointestinal symptoms in the 24 h after consuming the plant-based mince pasta meal. It is also important to note, however, that the type of fibre in the plant-based mince would be expected to influence the manifestation of adverse gut symptoms, with some fermentable fibres more likely to lead to bloating and gas production [10,11]. Therefore, a detailed analysis of fibre composition of plant-based meat products would be of interest. Previous studies have suggested that consuming plant-based meats over the longer term could potentially lead to favourable effects on the gut microbiome [12], which may have the potential to lead to improved gut health. To date, however, there have been no adequately powered or controlled studies that have directly assessed this, so further work is required to be able to draw robust conclusions. It is also important to note that this study was conducted in healthy male volunteers, and that individuals with significant gastrointestinal issues were excluded (given the potential impact of these conditions on appetite). As a result, it would be interesting to determine whether the results may have been different had we conducted the study in a cohort with poorer gut health.

### Strengths and Limitations

The strengths of this study relate to the randomised cross-over design, which ensured that individuals were providing with the plant-based and beef pasta meals in a random order and were blinded as to treatment order. The pasta meals were also carefully matched so that all ingredients, except for the mince, were consistent between recipes. The study was also conducted in a controlled environment, in which participants were provided with an identical breakfast meal prior to the lunch meal at each clinic appointment and remained in the clinic throughout the study to limit variations in physical activity or environment that could impact their appetite or post-meal satiety. The plant-based and beef pasta test meals and buffet meals were provided to participants in individual booths, so that intake could not be unduly influenced by observing the amounts that other participants consumed. We assessed satiety both via subjective scores at frequent intervals before and after the meal and via intake at the next (buffet) meal, increasing the strength of the conclusions.

While the study has many strengths, it is also important to acknowledge the limitations. It is particularly important to note that this study was conducted in a population of healthy, young males, and we cannot be certain that equivalent results would be obtained in other population groups. The volumes of food that were consumed were very high (~9000 kJ across the 2 meals that were served in the clinic), and it may be expected that the energy deficits would be lower in individuals who consumed lower volumes of food. Males were deliberately selected for this proof-of-concept study due to the added complexity of undertaking studies related to food intake in females, given the impact of the hormonal changes that occur across the menstrual cycle on appetite and satiety control. Nevertheless, repeating this study in a female cohort would be valuable to confirm whether similar results are obtained. It is also important to note that this study only examined a single meal event, and whether the same satiating effect would be maintained over multiple meals and days requires further investigation.

## 5. Conclusions

The results of this study support our hypothesis that individuals need to consume lower volumes and amounts of energy to achieve satiety when consuming a meal prepared with plant-based mince, compared to an equivalent meal prepared with beef mince. In addition, we found that measures of satiety and energy consumption at a subsequent buffet meal were not different following the plant based and beef meals, suggesting that the lower energy consumption of the plant-based pasta meal did not result in greater hunger post-prandially or a compensatory increase in energy consumption at the subsequent meal event. There were no adverse effects of consuming the high-fibre plant-based pasta meal on measures of gut comfort in this healthy male population. While these results are promising, further studies are required to establish the longer-term effects of replacing meat with plant-based mince on energy intakes, including the potential for these products to support weight maintenance and weight loss. The mechanisms underlying the lower consumption of the plant-based mince meal compared to the beef mince meal and the potential benefits of longer-term consumption of these products on the microbiome and gut health would also be valuable to explore.

## Figures and Tables

**Figure 1 nutrients-14-05326-f001:**
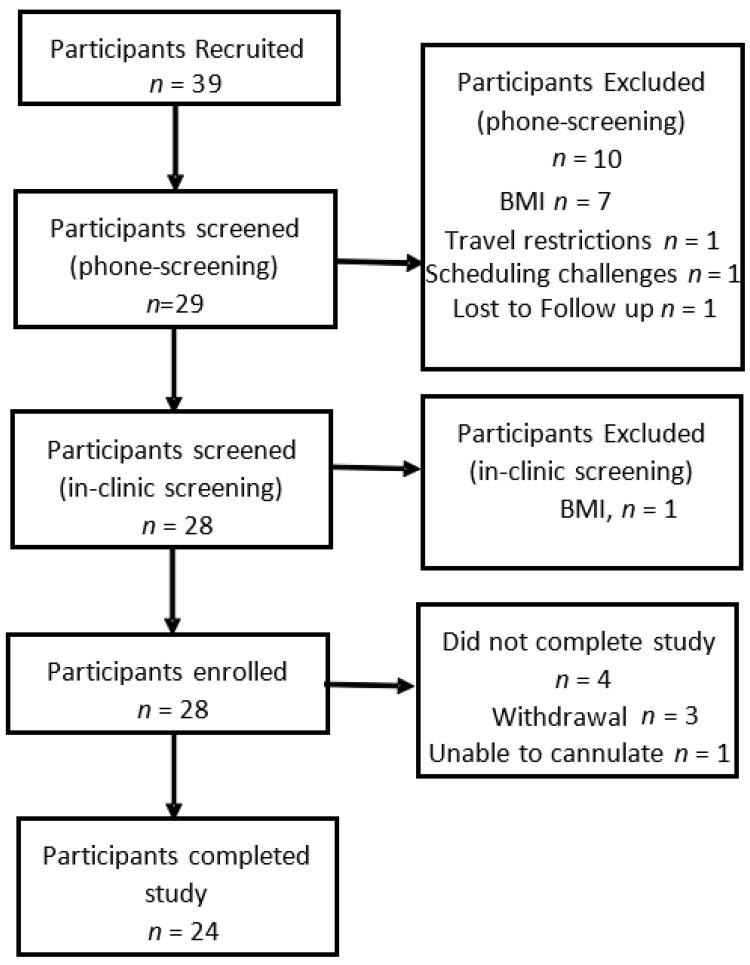
Participant Flow.

**Figure 2 nutrients-14-05326-f002:**
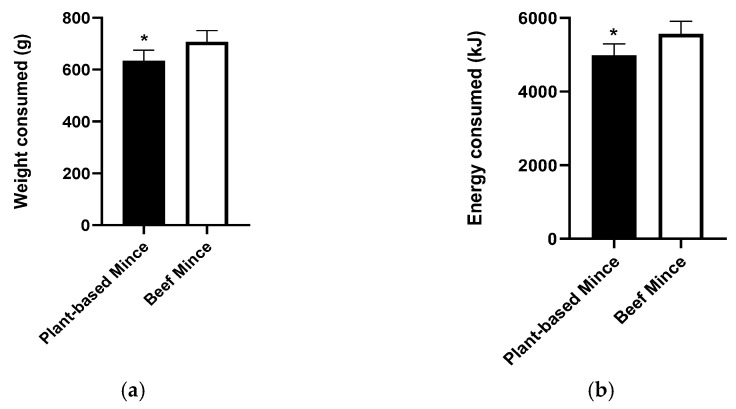
(**a**) Weight and (**b**) Energy consumption of the pasta lunch meal prepared with plant-based (filled histograms) and beef (open histogram) mince. Both weight and energy intake of the pasta lunch meal was lower for the meals prepared with plant-based mince compared to beef mince (*, paired *t*-test, *p* < 0.05).

**Figure 3 nutrients-14-05326-f003:**
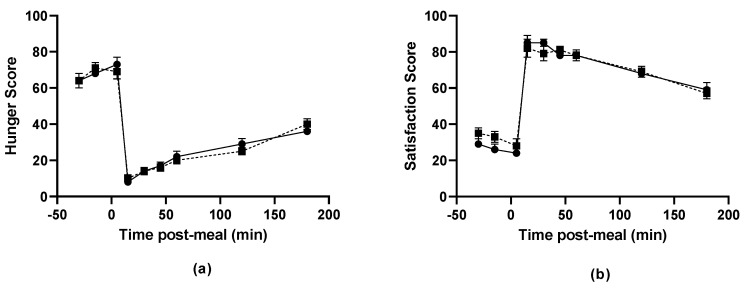
Measures of (**a**) Hunger, (**b**) Fullness, (**c**) Satisfaction and (**d**) Desire to eat, as assessed using a validated visual analogue scale (VAS) before and after consumption of the pasta lunch meal prepared with plant-based (closed circles and solid lines) and beef (closed squares and dotted lines) mince. There were no differences in any of these measures either before or after consumption of the test meals (repeated measures ANOVA, *p* < 0.05).

**Figure 4 nutrients-14-05326-f004:**
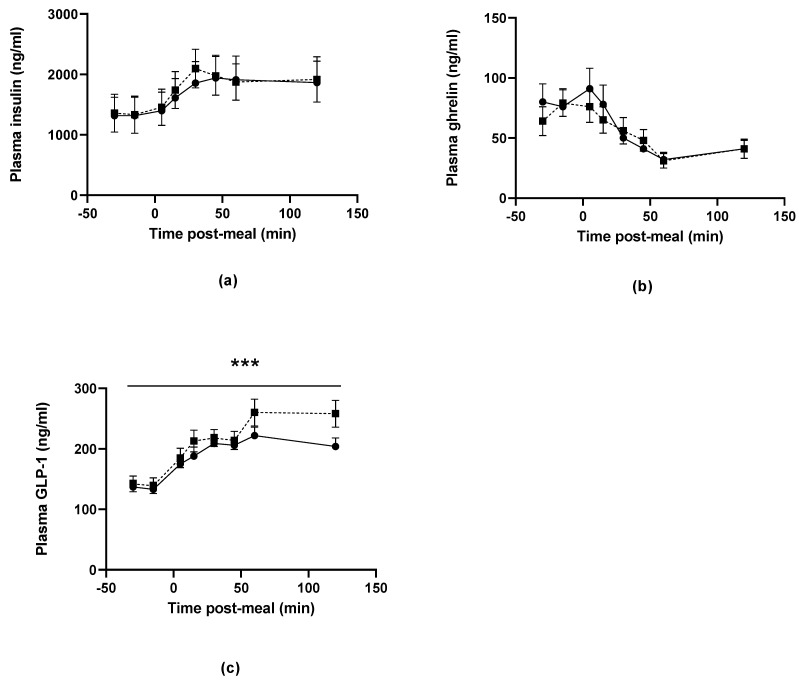
Plasma concentrations of (**a**) Insulin, (**b**) Ghrelin and (**c**) GLP-1 before and after consumption of the pasta lunch meal prepared with plant-based (closed circles and solid lines) and beef (closed squares and dotted lines) mince. There were no differences in the concentrations of insulin or GLP-1 between the plant-based and beef mince meals either before or after consumption, whereas GLP-1 concentrations were lower following the consumption of the plant-based mince compared to the beef mince (***, repeated measures ANOVA, F = 3.91; *p* < 0.001).

**Table 1 nutrients-14-05326-t001:** Nutritional composition of the plant based and beef pasta meals.

Nutritional Component	Plant-Based Mince Pasta Meal	Beef Mince Pasta Meal
Moisture (% weight)	55.0	59.8
Ash (% weight)	1.9	1.2
Protein (% weight)	12.6	12.3
Fat (% weight)	8.6	8.6
Starch (% weight)	14.0	14.8
Sugars (% weight)	1.8	1.5
Dietary Fibre (% weight)	4.8	2.1

**Table 2 nutrients-14-05326-t002:** Symptoms of gut health, as assessed by a validated gut health score questionnaire, in the 24 h before and 24 h after consuming the plant-based mince and beef mince pasta meals.

	Before Meal			24 h after Meal		
Symptom	Plant-Based Mince	Beef Mince	*t*-Test	Plant-Based Mince	Beef Mince	*t*-Test
Gut Pain	4.92 ± 0.06	4.96 ± 0.04	0.99	4.81 ± 0.09	4.59 ± 0.24	0.33
Upper gut fullness	4.08 ± 0.18	4.17 ± 0.18	0.99	3.86 ± 0.19	3.69 ± 0.27	0.49
Bloating	4.63 ± 0.13	4.92 ± 0.06	0.33	4.38 ± 0.16	4.37 ± 0.24	0.20
Excessive gas	4.71 ± 0.11	4.54 ± 0.13	0.63	4.52 ± 0.18	4.42 ± 0.24	0.33
Burping/Belching	4.92 ± 0.06	4.92 ± 0.06	0.67	4.81 ± 0.15	4.64 ± 0.23	0.99
Gurgling	4.67 ± 0.10	4.71 ± 0.11	1.00	4.62 ± 0.13	4.41 ± 0.25	0.58
Frequent bowel movements	4.58 ± 0.15	4.71 ± 0.11	0.72	4.67 ± 0.11	4.50 ± 0.24	0.45
Hunger	3.50 ± 0.12	3.38 ± 0.12	0.61	3.48 ± 0.15	3.42 ± 0.22	0.27
Trouble finishing meals	4.88 ± 0.09	5.00 ± 0.00	0.58	4.81 ± 0.11	4.55 ± 0.24	0.19
Regurgitation	5.00 ± 0.00	5.00 ± 0.00	0.33	5.00 ± 0.00	4.72 ± 0.23	0.99
Urgent bowel movements	4.88 ± 0.09	5.00 ± 0.00	1.00	4.95 ± 0.05	4.72 ± 0.23	0.19
Diarrhoea	4.96 ± 0.04	4.96 ± 0.04	0.33	4.86 ± 0.14	4.72 ± 0.22	0.99
Constipation	5.00 ± 0.00	5.00 ± 0.00	0.33	4.95 ± 0.05	4.77 ± 0.23	0.99
Nausea	5.00 ± 0.00	5.00 ± 0.00	0.33	4.95 ± 0.05	5.00 ± 0.00	0.99
Heartburn	4.96 ± 0.04	4.92 ± 0.06	0.58	4.90 ± 0.07	4.73 ± 023	0.33
Uncontrolled stools	5.00 ± 0.00	4.96 ± 0.04	0.33	4.90 ± 0.18	4.73 ± 0.23	0.33

## Data Availability

Data is available on request.

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
