# Peer review of "Assessing the Effect of Plant-Based Mince on Fullness and Post-Prandial Satiety in Healthy Male Subjects"

_nutrients, 2022, doi:10.3390/nu14245326_

Round 1
Reviewer 1 Report
Several critical comments needs to be addressed before the review can continue
Any reason why only male participants?
Table 1. I wonder if the caloric value differs? If yes, then how does the author counter the difference?
No power analysis for sample-size determination? Is 24 appropriate?
Can the authors add effect size for their significant results?
Section 3.4, please add the results of AUC as appendix/supplemetary
Author Response
We thank the reviewer for their comments, and provide responses to the specific queries below.
- Any reason why only male participants?
Males were deliberately selected for this proof-of-concept study due to the added complexity of undertaking studies related to food intake in females, given the impact of the hormonal changes that occur across the menstrual cycle on appetite and satiety control. We have acknowledged this as a limitation in the discussion section of the manuscript and indicated that repeating this study in a female cohort would be valuable to confirm whether similar results are obtained (Page 14, Line 442-447).
- Table 1. I wonder if the caloric value differs? If yes, then how does the author counter the difference?
The total caloric value of the pasta meal was similar between the plant-based and beef pasta meals (beef mince pasta meal, 788 kJ/100g; plant-based mince pasta meal, 785 kJ/100g), and this information has now been included in the revised manuscript (Page 4, Line 150-152). It should also be noted that any differences in the total caloric value of the pasta meals would not have affected the results of the study, since participants were provided with an amount of food that was substantially higher than what they were expected to be able to consume, and none of the participants consumed the entire pasta meal. Thus, we measured the amount of food that was remaining once the participants indicated that they were full and subtracted this from the weight of the pasta meal at the start and used this figure to calculate the actual amount (weight) and energy consumed.
- No power analysis for sample-size determination? Is 24 appropriate?
While there was limited data available to conduct a power analysis, we based the sample size for the current study on findings reported by Gregersen et al, 2008. This power analysis suggested that, using a paired-study design and a standardised diet, a sample size of 23 would provide 90% power to detect a 500kJ difference in energy intake for ad libitum meals following test meal consumption.
Gregersen NT, Flint A, Bitz C, Blundell JE, Raben A, Astrup A Reproducibility and power of ad libitum energy intake assessed by repeated single meals (2008)Am J Clin Nutr, 87(5):1277-1281.
- Can the authors add effect size for their significant results?
We have now added effect sizes for all significant results in the manuscript.
- Section 3.4, please add the results of AUC as appendix/supplementary
The AUC values for all these measures have now been provided in Section 3.4 (Page 8, Lines 272– 276). AUC values for the metabolic and gut hormones have also been added to Section 3.6
Reviewer 2 Report
This study supports that higher fiber and lower energy density may result in lower energy intake at a meal. The manuscript is very organized, thorough, and clearly written. Results are clear and the Discussion addresses results, strengths, and limitations.
Line 50: Is there a reference for this?
Figure 3. At the end of the note, it should say beef mince – plant mince is repeated.
Line 339: I suggest putting “male” in front of volunteers.
Author Response
REVIEWER 2 We thank the reviewer for describing the manuscript as organised, thorough, and clearly written. Responses to the specific comments raised are provided below. 1. Line 50: Is there a reference for this? This information has been obtained through personal communication from v2food, based on feedback received from consumers and staff, so there is no specific reference. Reference to this personal communication has been made in the revised manuscript (Page 2, Line 53). 2. Figure 3. At the end of the note, it should say beef mince – plant mince is repeated. This has been corrected 3. Line 339: I suggest putting “male” in front of volunteers. |
|
|
Must be improved |
Not applicable |
This has been amended (Page 12, Line 376).
Reviewer 3 Report
The article describes a randomised, single-blind, crossover trial involving 24 healthy male subjects to assess the effect of replacing minced beef with minced meat of vegetable origin in a pasta meal, both on the quantity of the meal consumed and on the feeling of satiety in the post-meal period. The evaluation was carried out by subjective tests as well as by objective tests (biochemical determinations) and by the amount consumed in a subsequent meal.
The subject matter of the study is of great interest, an interest that is well justified by the authors in the introduction. Likewise, the design of the study, in all aspects addressed, is very well established and perfectly described. The results and their discussion are also well described.
The limitations found in the study coincide with those established or suggested by the authors: low number of participants, all of the same sex and with the same health conditions, which may bias the results. However, given the originality (which may explain the scarcity of bibliographical references present in the article) and the design of the study.
Author Response
We thank the reviewer for their positive feedback on the manuscript, and for appreciating the novelty of our study. There are no specific queries that need to be addressed.